# Dietary Soy Prevents Alcohol-Mediated Neurocognitive Dysfunction and Associated Impairments in Brain Insulin Pathway Signaling in an Adolescent Rat Model

**DOI:** 10.3390/biom12050676

**Published:** 2022-05-08

**Authors:** Ming Tong, Jason L. Ziplow, Princess Mark, Suzanne M. de la Monte

**Affiliations:** 1Liver Research Center, Division of Gastroenterology, Department of Medicine, Rhode Island Hospital, Alpert Medical School of Brown University, Providence, RI 02808, USA; mtong216@gmail.com (M.T.); ziplowj@chop.edu (J.L.Z.); princess.mark-adjeli@nih.gov (P.M.); 2Liver Research Center, Division of Gastroenterology, Departments of Medicine, Neurology and Pathology and Laboratory Medicine, Rhode Island Hospital, Providence, RI 02808, USA; 3Women and Infants Hospital of Rhode Island, Alpert Medical School of Brown University, Providence VA Medical Center, Providence, RI 02808, USA

**Keywords:** alcohol, temporal lobe, adolescence, dietary soy, insulin sensitizer, Morris water maze, aspartyl-asparaginyl-β-hydroxylase, Notch, brain atrophy

## Abstract

Background: Alcohol-related brain degeneration is linked to cognitive-motor deficits and impaired signaling through insulin/insulin-like growth factor type 1 (IGF-1)-Akt pathways that regulate cell survival, plasticity, metabolism, and homeostasis. In addition, ethanol inhibits Aspartyl-asparaginyl-β-hydroxylase (ASPH), a downstream target of insulin/IGF-1-Akt signaling and an activator of Notch networks. Previous studies have suggested that early treatment with insulin sensitizers or dietary soy could reduce or prevent the long-term adverse effects of chronic ethanol feeding. Objective: The goal of this study was to assess the effects of substituting soy isolate for casein to prevent or reduce ethanol’s adverse effects on brain structure and function. Methods: Young adolescent male and female Long Evans were used in a 4-way model as follows: Control + Casein; Ethanol + Casein; Control + Soy; Ethanol + Soy; Control = 0% ethanol; Ethanol = 26% ethanol (caloric). Rats were fed isocaloric diets from 4 to 11 weeks of age. During the final experimental week, the Morris Water maze test was used to assess spatial learning (4 consecutive days), after which the brains were harvested to measure the temporal lobe expression of the total phospho-Akt pathway and downstream target proteins using multiplex bead-based enzyme-linked immunosorbent assays (ELISAs) and duplex ELISAs. Results: Ethanol inhibited spatial learning and reduced brain weight, insulin signaling through Akt, and the expression of ASPH when standard casein was provided as the protein source. The substitution of soy isolate for casein largely abrogated the adverse effects of chronic ethanol feeding. In contrast, Notch signaling protein expression was minimally altered by ethanol or soy isolate. Conclusions: These novel findings suggest that the insulin sensitizer properties of soy isolate may prevent some of the adverse effects that chronic ethanol exposure has on neurobehavioral function and insulin-regulated metabolic pathways in adolescent brains.

## 1. Introduction

Alcohol-related brain disease (ARBD), characterized by neurobehavioral abnormalities and cognitive-motor deficits [1], can progress to dementia and disability [2,3]. Brain atrophy, a consistent neuroanatomical substrate of ARBD [3,4,5,6,7,8,9,10,11,12,13,14], targets prefrontal, temporal, and cerebellar cortical and white matter structures and the corpus callosum [5,15]. Although the severity of ARBD correlates with maximum daily and lifetime consumption [5,10,16], the developmental stage is another critical variable such that the impact of heavy alcohol exposures is greater in immature compared with mature brains [5,17,18,19]. Moreover, adolescents are highly susceptible to ethanol-mediated sustained neurocognitive dysfunction [17].

Mechanistically, chronic and/or binge ethanol exposure inhibits brain insulin and insulin-like growth factor, type 1 (IGF-1) signaling through phosphoinositol-3-kinase (PI3K)-Akt pathways that regulate cell survival, gene and protein expression, growth, metabolism, and plasticity [20,21,22,23,24,25,26,27,28,29,30]. Aspartyl-asparaginyl-β-hydroxylase (ASPH; formerly abbreviated AAH), an ~86 kD transmembrane phosphoprotein expressed on cell surfaces and endoplasmic reticulum membranes [31,32,33], is stimulated by insulin/IGF-1-PI3K-Akt and inhibited by ethanol [21,34,35,36]. ASPH regulates Notch pathway signaling via the hydroxylation of Asp and Asn β-carbons in the EGF-like domains of Notch and Jagged [31,33]. In addition, ASPH signals through hypoxia-inducible factor 1 alpha (HIF-1α) to drive cell motility under conditions of oxidative stress [37,38,39]. Humbug, a highly related truncated protein ( ~37 kD) that shares most of its amino acid sequence with the N-terminal region of ASPH [32], is also regulated by insulin/IGF-1 and inhibited by ethanol [21,34,36], but due to its lack of a C-terminal catalytic domain, does not regulate Notch. Instead, Humbug modulates calcium flux within the ER [40] and cell adhesion [21,41]. Ethanol inhibition of insulin/IGF-1-PI3K-Akt-ASPH-Notch can broadly impact neuronal and glial functions across the lifespan [42,43] due to Notch’s multifaceted effects on nerve regeneration [44], cell cycle progression, neurodevelopment, histogenesis [45], cell motility, cell adhesion [31,38,41,46] and neuronal plasticity [47].

Given the complexity of metabolic abnormalities caused by excessive ethanol consumption, it may be advantageous to therapeutically target proximal points in the insulin/IGF-1 signaling cascade to broadly restore the function of downstream pathways. For example, peroxisome proliferator-activated receptor (PPAR) agonists that have insulin-sensitizing effects were demonstrated to restore insulin/IGF-1 signaling and reduce organ pathology in experimental rat models of chronic ethanol exposure [48,49,50,51]. In addition, PPAR agonist treatments were shown to normalize the neurobehavioral function or neuroglial expression of brain insulin/IGF-1 target genes and proteins in both ethanol and non-ethanol exposure models [48,52,53,54]. In a preliminary study designed to evaluate more accessible approaches for remediating the adverse CNS developmental effects of chronic ethanol exposure, we discovered that dietary soy, which has known insulin-sensitizer actions [55,56], could prevent both placental and craniofacial phenotypic pathologies in an experimental model of fetal alcohol spectrum disorder (FASD) [57].

The aim of this study was to test the hypothesis that chronic-ethanol-exposure-induced neurobehavioral abnormalities and associated impairments in brain insulin signaling through cell survival, metabolism, and neuronal plasticity pathways could be prevented by dietary soy isolate due to its potent insulin sensitizing effects. The study’s endpoints included assessments of neurocognitive functions, brain weight, and temporal lobe insulin/IGF-1 signaling through PI3K-Akt-ASPH-Notch. This study attempts to narrow gaps in knowledge about the potential use of widely accessible dietary/public health measures to reduce the rate and severity of alcohol-related brain disease/dysfunction. The approach was mechanistically driven and designed to characterize the impact of dietary soy on ethanol-mediated impairments in brain insulin signaling through growth, metabolic, and plasticity pathways [20,21,22,23,24,25,26,27,28,29,30]. The findings herein provide new information about the neuroprotective effects of dietary soy in an adolescent chronic ethanol exposure model.

## 2. Materials and Methods

### 2.1. General

The Institutional Animal Care and Use Committee (IACUC) at the Lifespan/Rhode Island Hospital approved the use of rats for this experiment. Our approved protocols followed guidelines established by the National Institutes of Health. The rats were housed under standardized humane conditions including a 12-hour light (7 a.m.–7 p.m.)/dark (7 p.m.–7 a.m.) cycle, controlled temperature (70 °F–74 °F), and with free access to food.

### 2.2. Experimental Model

Pregnant Long Evans rats were purchased from Harlan Sprague Dawley, Inc. (Indianapolis, IN, USA) and the pups were weaned at 3 weeks of age. Male and female offspring from seven different litters were randomly allocated to one of the four experimental sub-groups which were designated as follows: Control + Casein-CC; Ethanol + Casein-EC; Control + Soy-CS; Ethanol + Soy-ES. Twelve rats, comprising six males and six females, were included in each sub-group of this 4-way model. Beginning at 4 weeks of age, the rats were fed ad libitum with Lieber-DeCarli isocaloric liquid diets purchased from BioServ (Frenchtown, NJ, USA). Casein (41.4 kcal/L) is the standard protein source in the Lieber-DeCarli diet and therefore was regarded as the Control protein. Soy Isolate (41.4 kcal/L) replaced Casein as the experimental protein source. CC and CS diets contained 0% ethanol. EC and ES diets contained 37% ethanol (caloric) for the initial 3 weeks of feeding, and 26% ethanol (caloric) for the remainder of the study. This approach effectively generated a model of alcohol-related brain pathology while ensuring adequate weight gain over the course of the experiment [29,58]. Rats were monitored daily and weighed weekly to ensure adequate food intake and growth (Figure 1A,B). In contrast to previous studies [29,58], binge alcohol administration was not superimposed on chronic ethanol feeding.

### 2.3. Morris Water Maze

During the final experimental week (Week 7), the rats (11 weeks old) were subjected to the Morris Water Maze (MWM) test as described [29]. The circular maze measured 182.88 cm in diameter and 60.96 cm in height. The water, maintained at 24 °C and with a depth of 35 cm, was opacified with white non-toxic water-soluble paint. The platform, when submerged, was 1.0 cm below water level. The diets and ethanol feedings were continued throughout the experiment, including during the period of MWM testing to avoid causing alcohol withdrawal. Furthermore, previous studies demonstrated that 6 weeks of chronic ethanol feeding were sufficient to generate alcohol-related neurodegeneration [22]. MWM tests were performed between 10 a.m. and 4 p.m. (Day Cycle) and were conducted over four consecutive days with three trials per day. On Day 1, the rats learned to locate, swim to, and land on a visible platform in the center of the maze from different start points. On Days 2–4, the platform was submerged. On Day 2, the rats learned to find the hidden platform from a single-entry point. On Days 3 and 4, the maze entry quadrants were randomized for each trial. The rats were provided 120 seconds to locate the platform, after which they were guided and placed onto the platform to terminate the trial. Recordings of the trials were analyzed using the EthoVision 8.5 (Noldus, Leesburg, VA, USA) [54,59]. Speed and latency (seconds required for the rats to locate and land on the platform) were recorded for three daily trials and on four consecutive days. Inter-group comparisons were based on area-under-curve calculations corresponding to the latencies measured over the three daily trials per rat. The area under curve is an integrated measurement of an effect. Data were analyzed using two-way ANOVA with post hoc Tukey tests. All rats were fully awake and alert and exhibited no signs of intoxication at testing. Ample breaks (at least 10 min) were provided between trials.

### 2.4. Sacrifice

At the end of the experiment, under deep isoflurane anesthesia, the rats were sacrificed by cardiac puncture followed by exsanguination. The brains were harvested immediately, and a standardized 3 mm thick coronal slice that flanked the hypothalamus/infundibulum and included both temporal lobes was snap-frozen on dry ice and stored at −80 °C for molecular and biochemical assays.

### 2.5. Preparation of Protein Homogenates 

Temporal lobe samples were homogenized in 5 volumes of buffer containing 50 mM Tris (pH 7.5), 150 mM NaCl, 5 mM EDTA (pH 8.0), 50 mM NaF, 0.1% Triton X-100, and protease (1mM PMSF, 0.1 mM TPCK, 1 mg/mL aprotinin, 1 mg/mL pepstatin A, 0.5 mg/mL leupeptin, 1 mM NaF, 1 mM Na_4_P_2_O_7_) and phosphatase (2 mM Na_3_VO_4_) inhibitors using a TissueLyser II (Qiagen, Germantown, MD, USA) with 5 mm stainless steel beads. Supernatants obtained after centrifuging the samples at 14,000× *g* for 10 min at 4 °C were used for multiplex and duplex enzyme-linked immunosorbent assays (ELISAs). Protein concentrations were measured using the bicinchoninic acid (BCA) assay.

### 2.6. Multiplex Akt and Phospho-Akt Pathway ELISAs

We used commercial bead-based Total and Phospho Akt Magnetic 7-Plex Panels (Life Technologies Corp, Frederick, MD, USA) to measure the effects of ethanol and dietary soy on the expression and phosphorylation of proteins integrally related to insulin (IN) and IGF-1 signaling through Akt and glycogen synthase kinase 3β (GSK-3β). The Total-Akt panel measured immunoreactivity to the insulin receptor (IN-R), IGF-1R, insulin receptor substrate, type 1 (IRS-1), Akt, glycogen synthase kinase 3β (GSK-3β), Ribosomal protein S6 kinase beta-1 (p70S6K), and proline-rich AKT substrate of 40 kDa (PRAS40). The Phospho-Akt panel measured immunoreactivity to ^pYpY1162/1163^-IR, ^pYpY1135/1136^-IGF-1R, ^pS312^-IRS-1, ^pS473^-Akt, ^pS9^-GSK-3β, ^pT246^-PRAS40, and ^pTpS421/424^-p70S6K according to the manufacturer’s protocol. In brief, brain tissue homogenates (100 µg protein in 50 µl of lysis buffer) were incubated with antibody-bound beads. Captured antigens were detected with biotinylated secondary antibody and phycoerythrin-conjugated Streptavidin. Fluorescence intensity was measured in a MAGPIX (Bio-Rad, Hercules, CA, USA). Data are expressed as fluorescence light units (FLU) and were corrected for protein concentration.

### 2.7. Duplex Enzyme-Linked Immunosorbent Assays (ELISAs)

Direct binding duplex ELISAs were used to measure ASPH, Humbug, Notch 1, Jagged 1, HES-1, and HIF-1α protein expression. Target protein immunoreactivity was detected with unique unlabeled monoclonal antibodies, horseradish peroxidase-conjugated secondary antibody, and Amplex UltraRed soluble fluorophore (Invitrogen, Carlsbad, CA, USA), as described [20,60]. To adjust for variability in sample loading, the results were normalized to large acidic ribosomal protein (RPLPO) (Proteintech Group Inc, Chicago, IL, USA) [48,61] which was biotinylated and detected with streptavidin-conjugated alkaline phosphatase and the 4-Methylumbelliferyl phosphate (4-MUP) substrate. Fluorescence intensities (Amplex Red: Ex 560 nm/Em 590 nm; 4-MUP: Ex360/Em450) were measured in a SpectraMax M5 (Molecular Devices, Sunnyvale, CA, USA). Antibody omission controls were included. The calculated target protein/RPLPO ratios were used for our statistical analysis.

### 2.8. Statistics

Inter-group comparisons were made using an analysis of variance (ANOVA) with Tukey post hoc tests (GraphPad Prism 9, San Diego, CA, USA). Significant (*p* < 0.05) and trend-wise (0.05 < *p* < 0.10) post hoc test differences are shown in the graphs and Tables.

### 2.9. Materials

Pharmaceutical grade ethanol was used in the in vivo experiments. The A85G6 and A85E6 monoclonal antibodies to ASPH and Humbug were generated into human recombinant proteins and purified over Protein G columns (Healthcare, Piscataway, NJ, USA) [34]. All other primary antibodies used for the duplex ELISAs were purchased from Abcam (Cambridge, MA, USA). RPLPO antibody was sourced from the Proteintech Group Inc (Chicago, IL, USA). ELISA MaxiSorp 96-well plates were purchased from Nunc (Rochester, NY, USA). Horseradish peroxidase (HRP)-conjugated secondary antibody, and Amplex Red soluble fluorophore were purchased from Invitrogen (Carlsbad, CA, USA). The SpectraMax M5 microplate reader was purchased from Molecular Devices Corp. (Sunnyvale, CA, USA). BCA reagents were obtained from Pierce Chemical Corp. (Rockford, IL, USA). All other fine chemicals were purchased from CalBiochem (Carlsbad, CA, USA), Pierce (Rockford, IL, USA), or Sigma (St. Louis, MO, USA).

## 3. Results

### 3.1. Effects of Ethanol, Sex, and Dietary Soy on Body Weight and Brain Weight

Over the 7-week course of the experiment, all rats progressively gained weight (Figure 1A,B). Among the rats fed with liquid diets that contained casein as the protein source, ethanol exposure and female sex were initially associated with lower mean body weights relative to the corresponding males, but during the latter half of the experiment, both the control and ethanol-exposed females had significantly lower mean body weights than males (Figure 1A). Dietary soy compressed the inter-group differences in mean body weight during the first 3 to 4 weeks, but during the latter half of the experiment, control and ethanol-exposed females weighed significantly less than corresponding males (Figure 1B). A two-way ANOVA demonstrated significant sex, exposure (ethanol and dietary soy), and sex x exposure interactive effects on body weight (Table 1). The heatmap of time-dependent changes in body weight stratified by sex, dietary protein source, and ethanol exposure depicts consistently lower mean body weights linked to female sex and ethanol, and ultimately reveals higher body weights associated with soy versus casein (Figure 1C; post hoc Tukey tests).

To further examine the effects of dietary soy on growth, the area-under-curve for body weight over the course of the experiment was calculated for each rat and inter-group differences were evaluated with a two-way ANOVA (Figure 1D). In addition to confirming the significant inhibitory effects of female sex and ethanol exposure, the results showed overall positive effects of soy relative to casein, and a normalization of body weight in Ethanol + Soy males relative to Control + Casein males. In contrast, no such protective effects were observed in the Ethanol + Soy female group.

The two-way ANOVA test demonstrated significant sex and exposure (ethanol and dietary soy), but no sex x exposure interactive effects on terminal brain weight (Table 1). Within the casein or soy group, both control and ethanol-exposed females had significantly lower mean brain weights than the corresponding males (Figure 1E). Within-sex comparisons revealed ethanol-associated reductions in mean brain weight whether casein or soy was used as the protein source (Figure 1E). However, dietary soy positively impacted brain weight in the control and ethanol-exposed males, and control but not ethanol-exposed females. Consequently, the mean M-CS and F-CS brain weights were highest within sex, and in contrast to M-EC, the M-ES brain weight did not significantly differ from that observed in M-CC rats (Figure 1E,F). Moreover, although females generally had lower mean brain weights than males (Figure 1F), the positive effects of dietary soy in control females increased their mean brain weight, rendering it comparable to that of M-CC rats and higher than M-EC rats (*p* = 0.036). On the other hand, dietary soy failed to increase the brain weight of F-ES rats relative to that of the F-EC group (Figure 1F). In essence, dietary soy prevented ethanol-associated brain atrophy in males but not females.

### 3.2. Ethanol and Soy Effects on Morris Water Maze (MWM) Performance

Our initial analysis demonstrated no significant sex-dependent differences in MWM performance. Therefore, the data were combined to focus on the effects of ethanol and dietary soy. The mean (±S.D.) area-under-curve calculations corresponding to the latencies required to reach and land on the platform over the three daily trials are depicted in Figure 2A. The data were analyzed with a two-way ANOVA (Table 2) with post hoc Tukey tests (Figure 2). Significant inter-group differences were detected on Days 1, 3, and 4. On Day 1, both CS (*p* = 0.01) and ES (*p* = 0.03) rats performed significantly better (shorter mean latencies) than EC rats. On Day 2, performance was similar across all four groups. On Day 3, the mean latencies were significantly shorter, i.e., performance was better in ES versus EC (*p* = 0.047), and the mean latency of EC did not differ significantly from those of CC or CS, i.e., performance was normalized. On Day 4, the mean latency for the EC group remained longer than for CS (*p* = 0.01) and ES (*p* = 0.066) rats, and CS exhibited a positive statistical trend for better performance relative to CC (*p* = 0.10). In contrast, no significant inter-group differences were detected with respect to swim velocities (Figure 2B and Table 2).

In summary, MWM performance improved over time in all groups, and dietary soy, irrespective of ethanol exposure, significantly enhanced performance in the MWM test.

### 3.3. Ethanol and Dietary Soy Effects on Akt-Pathway Proteins

The initial analyses demonstrated no significant sex-dependent differences in temporal lobe insulin/IGF-1-Akt pathway protein or phosphoprotein expression. Therefore, the male and female data were combined to focus on the effects of ethanol and dietary soy. Insulin Receptor (IN-R), IGF-1 receptor (IGF-1R), insulin receptor substrate-type 1 (IRS-1), Akt, GSK-3β, PRAS40 and p70S6K proteins were measured with the Total Akt Magnetic 7-Plex Panel and analyzed with a two-way ANOVA (Table 3) and post hoc Tukey tests.

Significant inter-group differences or statistical trends were observed for IN-R, IRS-1, Akt, GSK-3β and p70S6K (Table 3). None of the inter-group differences were specifically driven by ethanol. Instead, they were either a result of a dietary protein effect (IN-R, GSK-3β), or ethanol–dietary protein interactive effects (IN-R, IGF-1R, IRS-1, Akt, GSK-3β, and P70S6K). The graphs with post hoc statistical comparisons show dominant ethanol + casein inhibitory effects such that IN-R (Figure 3A), IRS-1 (Figure 3C), Akt (Figure 3D), GSK-3β (Figure 3E), and P70S6K (Figure 3G) expression levels were significantly reduced relative to CC and/or ES. In contrast, EC had no observed effects on IGF-1R (Figure 3B) or PRAS40 (Figure 3F) expression. Of note is that dietary soy abrogated EC’s inhibitory effects on the insulin/Akt pathway protein expression.

### 3.4. Ethanol and Dietary Soy Effects on Phospho-Akt-Pathway Proteins

The Phospho-Akt Magnetic 7-Plex Panel was used to measure ^pYpY1162/1163^-IR, ^pYpY1135/1136^-IGF-1R, ^pS312^-IRS-1, ^pS473^-Akt, ^pS9^-GSK-3β, ^pT246^-PRAS40, and ^pTpS421/424^-p70S6K. Data were analyzed by two-way ANOVA (Table 4) with post hoc Tukey tests and results are depicted graphically in Figure 4. Significant effects of dietary protein (soy) were observed for pIN-R (*p* = 0.037), pIRS-1 (*p* = 0.041), and pPRAS40 (*p* = 0.004), and significant trends for ethanol x dietary protein interactive effects were observed for pIN-R (*p* = 0.089), pAkt (*p* = 0.006) and pGSK-3β (*p* = 0.008) (Table 4). The graphed results with post hoc statistical comparisons demonstrate a significantly higher expression of pIN-R (Figure 4A), pAkt (Figure 4D), and pGSK-3β (Figure 4E) in ES versus EC, reflecting a protective effect of dietary soy vis-a-vis chronic ethanol feeding. In addition, pAkt was significantly reduced in EC relative to CC (Figure 4D), indicating an inhibitory effect of ethanol with casein as the dietary protein source. The increased expression of pGSK-3β in ES compared with EC would have further supported the activation (phosphorylation) of Akt since S9 phosphorylation inhibits GSK-3β, blocking its inhibitory effects on Akt. Furthermore, pPRAS40 expression was selectively reduced in CS relative to CC and EC (Figure 4F). There were no significant effects of ethanol or dietary soy on the mean levels of pIGF-1R (Figure 4B), pIRS-1 (Figure 4C), or pP70S6K (Figure 4G).

### 3.5. Ethanol and Soy Effects on ASPH Proteins

A85G6-ASPH specifically binds to the C-terminal region of ASPH which contains the critical catalytic domain for Notch and Jagged hydroxylation [34]. A85E6-ASPH binds to the N-terminus of both the ~86 kD ASPH and the ~37 kD Humbug [34]; Humbug lacks the C-terminus of ASPH, but otherwise the two proteins are virtually identical [31,32,33,40,62]. Since previous studies have demonstrated that ASPH and Humbug were stimulated by insulin but inhibited by ethanol [21,34,36], it was of interest to determine how dietary soy impacted the expression of these proteins, as well as downstream signaling through Notch and HIF-1α. Duplex ELISAs were used to measure A85G6-ASPH, A85E6-ASPH, Notch-1, Jagged-1, HES-1, and HIF-1α. The results were normalized to RPLPO [48,61].

Since sex differences were not detected, data from males and females were analyzed together using two-way ANOVA tests (Table 5) with post hoc Tukey tests and are depicted graphically in Figure 5 and Figure 6. Significant ethanol or dietary soy effects occurred with respect to ASPH-A85G6 and ASPH-A85E6 (Table 5). In contrast, there were no significant ethanol x soy interactive effects on the expression levels of these proteins (Table 4). In rats fed with Casein as the dietary protein source, A85G6-ASPH expression was significantly reduced by chronic ethanol exposure, whereas among dietary soy-fed rats, there was no significant difference between the CS and ES groups (Figure 5A). Ethanol reduced the expression of A85E6-ASPH (ASPH + Humbug) relative to the control in both the casein and soy protein-fed groups (Figure 5B). However, a rescue effect of dietary soy occurred as evidenced by the significantly higher expression of ASPH-A85E6 in ES compared with EC.

### 3.6. Ethanol and Soy Effects on NOTCH Pathway Proteins

Our analysis of ethanol and dietary soy effects on the Notch pathway and HIF-1α signaling using a two-way ANOVA with post hoc Tukey tests revealed a significant dietary soy effect and an ethanol trend effect on Notch-1, and a significant ethanol effect on Jagged-1 (Table 4). In contrast, there were no significant or trend effects of ethanol-, soy, or ethanol x soy interactions with respect to HES-1 or HIF-1α. Post hoc Tukey tests demonstrated no significant pair-wise differences in the mean expression levels of Notch-1, Jagged-1, HES-1, or HIF-1α between the control and ethanol groups maintained with either dietary casein or soy (Figure 6). Instead, the main effects were significant increases in Notch-1 expression in CS relative to CC and EC (Figure 6A), and in Jagged-1 in CS versus EC (Figure 6B). HES-1 and HIF-1α were not significantly modulated by dietary soy or ethanol (Figure 6C,D).

## 4. Discussion

In this study, young adolescent male and female Long Evans rats were chronically fed control or ethanol-containing isocaloric liquid diets for 7 weeks to determine how the substitution of soy isolate for casein impacted body weight, brain weight, cognitive-behavioral function, and the temporal lobe expression of insulin/IGF-1/Akt pathway signaling molecules, ASPH, Notch network, and HIF-1α proteins. The underlying premise was that ethanol’s inhibitory effects on insulin/IGF-1 signaling through Akt in the brain lead to neurobehavioral and neurodevelopmental dysfunctions with a reduced expression of ASPH, Notch network proteins and HIF-1α [34,60]. Furthermore, several previous studies have shown that treatment with peroxisome-proliferator activated receptor (PPAR) agonists could restore impairments in insulin/IGF-1 signaling through Akt and downstream targets including ASPH [48,50,53,54]. With the awareness that PPAR agonists can have significant off-target adverse effects [63,64], together with evidence that soy protein and isoflavones positively impact insulin resistance [55,56,65] and could potentially prevent FASD [57], we designed an experiment to evaluate the therapeutic or preventive effects of soy isolate on brain pathology in an established model of chronic ethanol feeding. The model utilized 26% rather than 37% caloric ethanol in Liber–DeCarli liquid diets and excluded superimposed binge exposures because preliminary studies suggested that adolescent females could not tolerate maximum ethanol plus binge exposures (chronic + binge model) for the full 7-week duration of the experiment, yet an important requirement of NIH-funded research is to include both sexes. Therefore, we utilized a 4-way model in which the isocaloric control or ethanol-containing diets included either casein (standard) or soy isolate (experimental) as the sole source of dietary protein, and the ethanol dosing was moderately high and chronic. Although not as robust as the chronic + binge models used in other studies [60,66,67], the experimental approach ensured good health with continuous weight gain and the avoidance of seizure activity or lethargy in both male and female adolescent rats.

Males, as expected, consistently had higher mean body weights than females and chronic ethanol feeding with either casein or soy as the protein source reduced body weight in males and females. The potential mechanisms of ethanol’s inhibitory effects on growth include the suppression of the adolescent hypothalamic-pituitary-gonadal axis with attendant reductions in growth hormones and IGF-1 and delays in puberty [68]. On the other hand, soy, which is known to alter the function of the hypothalamic-pituitary axis [69], increased body weight relative to casein in both males and females.

The significantly lower mean brain weight in females compared with males was expected and consistent with what is already known about sex differences in brain weight [70]. Similarly, the growth-inhibitory or atrophying effects of ethanol on the brain have been well documented in humans [71,72] and experimental animals [73]. The substitution of soy isolate for casein modestly increased brain weight in male and female control rats, and reduced the brain atrophying effects of ethanol in ES relative to CC males. No such response occurred in females. Soy isolate’s normalization of brain weight in ethanol-exposed males (relative to CC) but not females suggests that soy may impart sex-selective neuroprotective effects during adolescence.

MWM testing revealed significant effects of dietary soy but not sex or ethanol. The failure to detect significant effects of ethanol contrasts with previous reports from our group [29]. It is likely that the difference in experimental outcomes was due to the less robust model generated by chronic compared with chronic + binge ethanol exposures. Nonetheless, there were two novel findings: (1) despite lower brain weights, female performance did not differ significantly from that of males; and (2) dietary soy generally improved MWM performance in both control and ethanol-exposed rats. Additional studies are needed to fully understand the translational significance of dietary soy’s positive effects on neurobehavioral function, even in the context of chronic ethanol feeding.

Previous studies have shown that chronic ethanol feeding impairs brain insulin signaling through the insulin receptor, IRS-1 and Akt pathways in both developing and mature brains [20,22,25,26,74,75]. However, casein was the standard protein included in the Lieber DeCarli liquid diets. To determine if the soy isolate-associated improvements in MWM performance were mediated by enhanced insulin/IGF-1 signaling through Akt, we used commercial multiplex panels to measure temporal lobe levels of both the total major pathway and phosphorylated signaling proteins. The main finding was that when casein was used as the protein source, chronic ethanol feeding impaired insulin/IGF-1 signaling through the insulin receptor, Akt, and pAkt as previously reported [20,22,25,26,74,75], but when soy isolate replaced casein vis à vis chronic ethanol feeding (ES), insulin receptor, Akt and the corresponding phosphorylated proteins were expressed at levels comparable to those measured in Casein control brains. Within the Casein + ethanol group, the significant reductions in temporal lobe IN-R and Akt reflect compromised signaling through pathways that regulate cell survival, neuronal plasticity, cell growth, energy metabolism and homeostasis [76,77,78]. In contrast, the normalization of IN-R, IRS-1, Akt, GSK-3β, and p70S6K expression levels in Soy + Ethanol (ES) brains likely represents an adaptive central nervous system (CNS) response to dietary soy isolate, which is known to support insulin signaling [55,56], and corresponds with improved MWM performance.

In addition to increasing the expression of IN-R-Akt pathway proteins, soy isolate enhanced the Ser9-induced phosphorylation of GSK-3β, which inactivates the kinase and prevents it from inhibiting Akt [79,80]. The inhibitory effect of soy isolate on pPRAS40 in control brains is noteworthy because the Proline-rich Akt substrate of 40 kDa (PRAS40) acts at the intersection of the Akt- and mammalian target of rapamycin (mTOR)-mediated signaling, and its phosphorylation contributes to the activation of the PI3K-Akt-mTOR network [81]. In addition, PRAS40 is a component of mTOR complex 1 (mTORC1). Evidence suggests that the links between the Akt and mTOR pathways may be important for reducing brain injury in the context of stroke [82]. On the other hand, a key component of mTOR activation is an increased expression and phosphorylation of P70S6K [82]. Dietary soy enhanced P70S6K protein expression only in ethanol-fed rats, which may have served to support the activation of mTOR without altering the constitutive phosphorylation of P70S6K. Additional studies are needed to evaluate the impact of soy isolate on signaling through mTOR in control and ethanol-exposed models.

Both ASPH-A85G6 and ASPH-A85E6, which, respectively, correspond to the ~86 kD full length protein and ~37 kD truncated protein, were significantly inhibited by ethanol + casein relative to the controls. In contrast, soy normalized ASPH-A85G6 and significantly increased ASPH-A85E6 in ethanol-fed rats. Since both ASPH-A85G6 and ASPH-A85E6 are stimulated by insulin signaling through Akt [34,83] and inhibited by ethanol [21,34], soy’s rescue effects on insulin signaling through Akt correspond with the normalized levels of ASPH in ethanol + soy relative to the control + Casein temporal lobe samples. Most likely, the stimulatory supportive effects of dietary soy on ASPH expression were due to preservation of signaling through IN-R and Akt rather than increased levels of peripheral blood insulin, since a follow-up analysis of serum insulin levels using a commercial magnetic bead-based Diabetes Bio-Plex Panel (Bio-Rad Laboratories, Hercules, CA, USA) showed similar (not statistically significantly different) levels across all four study groups (data not shown). This result was not surprising since soy isolate functions as an insulin sensitizer and anti-oxidant/anti-inflammatory agent, reducing insulin resistance rather than insulin production.

Although ASPH’s functions have mainly been linked to cell motility [31,33,41,83,84,85,86,87,88,89], calcium flux, and cell adhesion [40,41], growing evidence suggests that ASPH also inhibits cellular senescence [37]. Therefore, ethanol-associated brain atrophy and degeneration may be mediated in part by the inhibition of insulin-Akt stimulation of ASPH and attendant CNS cellular senescence and apoptosis. The supportive or rescue effects of dietary soy on these downstream target proteins are exciting because they link the restoration of insulin signaling to the preservation of adolescent brain structure and function.

The ~86 kD ASPH protein mediates its effects through Notch and Jagged [38,39,41] which have a consensus sequence for ASPH hydroxylation [33]. Notch signaling plays key roles in a broad range of neuronal and glial functions [42,43]. For example, impairments in Notch inhibit nerve regeneration [44]. The inhibition of the HES-1 transcription factor downstream of Notch reduces the expression of target genes that regulate the cell cycle [45], and disrupts neurodevelopment and histogenesis [45]. To determine how ethanol and dietary soy impact Notch networks in the adolescent temporal lobe, we measured Notch-1, Jagged-1, HES-1, and HIF-1α immunoreactivity with duplex ELISA whereby the results were normalized to RPLPO.

In contrast to previous observations in chronic ethanol-exposed postnatal developing brains [60] or adult livers [90], Notch pathway proteins and HIF-1α were not significantly inhibited by ethanol + casein. However, soy isolate was found to up-regulate Notch-1 expression in the controls relative to control + casein and ethanol + casein, and Jagged-1 relative to ethanol + casein, suggesting that dietary soy may have a positive, albeit modest stimulatory effect on Notch pathway signaling. Potential explanations for the discordant lack inhibitory effects of ethanol on Notch pathway proteins in the present study are as follows: (1) previous experiments utilized higher concentrations of ethanol (36% versus 26%) [34]; (2) the adult models were generated by combining repeated binge (2 g/kg, 3 x/week) with chronic exposures [60]; and (3) CNS Notch pathway vulnerability to ethanol varies with developmental stage, e.g., pre- and early post-natal [75] versus adolescent exposures. Additional studies are needed to further explore these responses in greater detail, including assessments of alternative Notch, Jagged and HES isoforms that may have greater importance in the adolescent brain.

## 5. Conclusions

The findings herein provide new information about the neuroprotective effects of dietary soy in an adolescent chronic ethanol-exposure model. Significant outcomes included improved neurobehavioral function, the enhanced expression of proteins critical to brain insulin signaling through metabolic, growth, cell survival, and plasticity pathways, and the normalized expression of ASPH downstream target proteins. A potentially exciting translational outcome of this research is that the neurocognitive effects of chronic ethanol exposure during adolescence may be prevented by incorporating soy isolate into the diet. Additional research is needed to determine the minimum amount of soy required to achieve these effects. Going forward, it is important to appreciate the complexity of soy isolate because, in addition to its potent insulin-sensitizing and anti-inflammatory effects [55,65,91], soy isolate contains significant amounts of choline [92,93] which has already been shown to be of benefit to neurocognitive functions following developmental exposures to alcohol, both in experimental models [94,95,96] and humans [97,98].

## Figures and Tables

**Figure 1 biomolecules-12-00676-f001:**
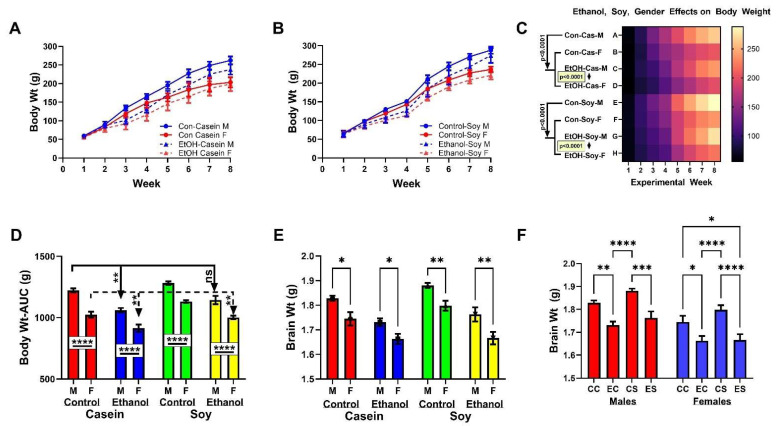
Effects of chronic ethanol and dietary soy isolate on body growth and brain weight in male and female adolescent Long Evans rats. A four-way model was generated with 12 rats per group (6 male, 6 female): Control + Casein, Ethanol + Casein, Control + Soy, Ethanol + Soy. From 4-weeks of age, rats were maintained for 7 weeks on isocaloric Lieber–Decarli liquid diets containing 0% or 37% (3 weeks) followed by 26% (4 weeks) (caloric) ethanol and with casein (standard) or soy isolate (experimental) protein. Weekly body weights were measured to compare sex and ethanol effects with (**A**) casein or (**B**) soy as the dietary source of protein. (**C**) Heatmap comparisons of body weight changes over time for all groups showing consistently lower mean weights with female sex and ethanol-exposure, and increased body weight in soy compared with corresponding casein groups. (**D**) Area-under-curve body-weight calculations over the 7-week study confirming results in Panel C and showing soy’s protective effects in ethanol-fed males but not females relative to casein controls. (**E**,**F**) Brains were weighed at sacrifice. Mean brain weights were consistently lower in females. Dietary soy had a modest positive effect on brain weight in controls and a significant effect in ethanol-fed males but not females. Graphed data depict mean ± S.D. Results were analyzed by two-way ANOVA (see Table 1) with post hoc Tukey tests [* *p* < 0.05; ** *p* < 0.01; *** *p* < 0.001; **** *p* < 0.0001; ns = not significant].

**Figure 2 biomolecules-12-00676-f002:**
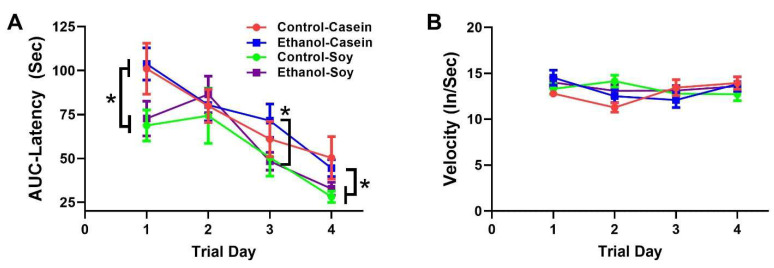
Effects of chronic ethanol and dietary soy isolate on Morris Water Maze (MWM) performance. MWM tests were run during the final week (Week 7) of the experiment, over 4 consecutive days with 3 trials per day per rat. Latencies to locate and land on the platform and swim velocities were video tracked and recorded, and results were analyzed using Ethovision 8.5. (**A**) Calculated area-under-curve data for latencies measured over the three daily trials were used for inter-group comparisons. (**B**) Swim velocities averaged over the 3 daily trials per rat were used for inter-group comparisons. Data from male and female rats were combined since sex differences were not observed. The two-way ANOVA test demonstrated significant effects of Trial Day and treatment (ethanol diet and soy isolate) on latency but not velocity (Table 2). In Panel A, the graphed data (mean ± S.D.) show progressive declines in latency for all groups, and significant effects of dietary soy isolate for reducing mean AUC latencies on Trial Days 1, 3, and 4 (* *p* < 0.05). Panel B shows no significant time-course or experimental group effects on velocity.

**Figure 3 biomolecules-12-00676-f003:**
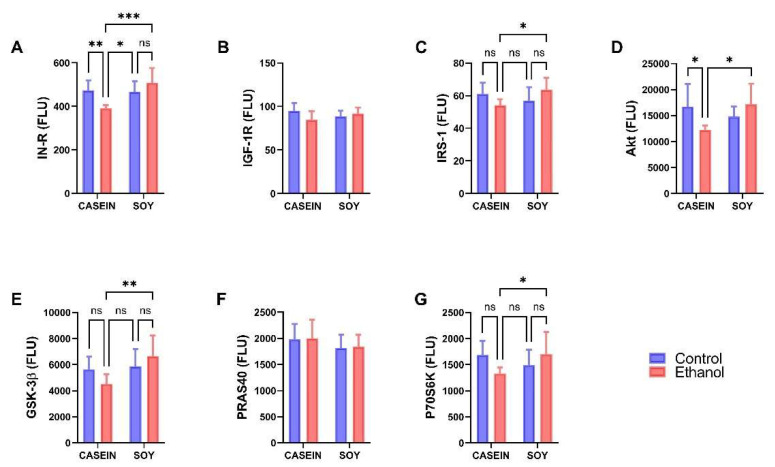
Effects of chronic ethanol and dietary soy isolate on insulin-IRS-1-Akt pathway protein expression in temporal lobe. Long Evans male and female young adolescent rats were fed for 7 weeks with isocaloric liquid diets containing 0% or 37% (3 weeks) followed by 26% (4 weeks) ethanol and casein (standard) or soy isolate as the protein source. Temporal lobe homogenates were used to measure (**A**) IN-R, (**B**) IGF-1R, (**C**) IRS-1, (**D**) Akt, (**E**) GSK-3β, (**F**) PRAS40 and (**G**) p70S6K with the Total Akt Magnetic 7-Plex Panel. Data from the male and female rats were combined since there were no sex-related differences. Graphed results reflect the mean ± S.D. of immunoreactivity expressed as fluorescence light units (FLU). Results were analyzed using two-way ANOVA tests (Table 3) with post hoc Tukey tests. Significant P-Values are indicated above the bars. * *p* < 0.05; ** *p* < 0.01; *** *p* < 0.001; NS = not significant.

**Figure 4 biomolecules-12-00676-f004:**
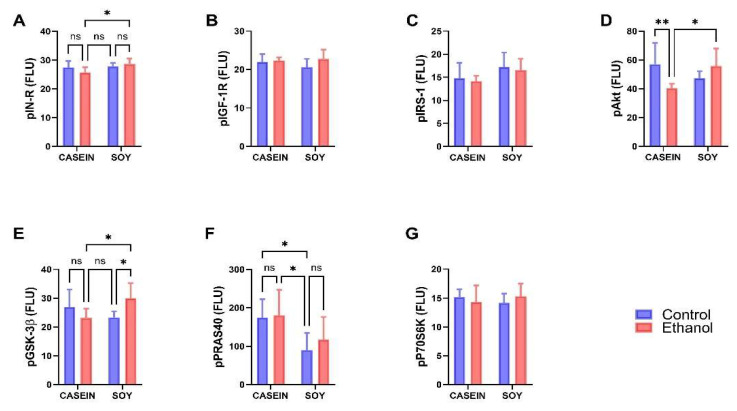
Effects of chronic ethanol and dietary soy isolate on insulin-IRS-1-Akt pathway phospho-protein expression in temporal lobe. Long Evans male and female young adolescent rats were fed for 7 weeks with isocaloric liquid diets containing 0% or 37% (3 weeks) followed by 26% (4 weeks) ethanol and casein (standard) or soy isolate as the protein source. Temporal lobe homogenates were used to measure (**A**) ^pYpY1162/1163^-IN-R, (**B**) ^pYpY1135/1136^-IGF-1R, (**C**) ^pS312^-IRS-1, (**D**) ^pS473^-Akt, (**E**) ^pS9^-GSK-3β, (**F**) ^pT246^-PRAS40, and (**G**) ^pTpS421/424^-p70S6K with the Phospho Akt Magnetic 7-Plex Panel. Data from the male and female rats were combined since there were no sex-related differences. Graphed data reflect the mean ± S.D. of immunoreactivity expressed as fluorescence light units (FLU). Results were analyzed with two-way ANOVA tests (Table 4) with post hoc Tukey tests. Significant *p*-Values are indicated above the bars. * *p* < 0.05; ** *p* < 0.01; ns = not significant.

**Figure 5 biomolecules-12-00676-f005:**
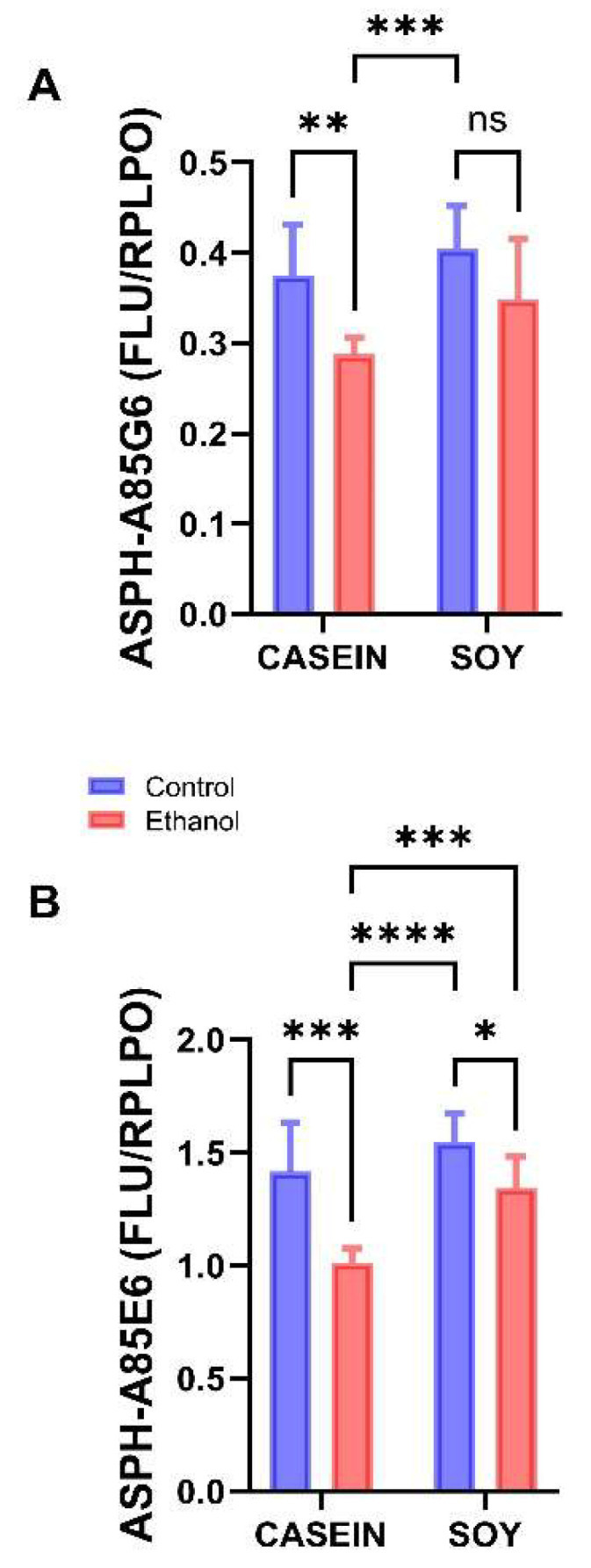
Effects of chronic ethanol and dietary soy isolate on ASPH protein expression. Duplex ELISAs were used to measure (**A**) ASPH using the A85G5 monoclonal antibody, and (**B**) Humbug + ASPH using the A85E6 monoclonal antibody. Immunoreactivity was detected with horseradish peroxidase-conjugated secondary antibody and Amplex UltraRed soluble fluorophore. Results were normalized to large acidic ribosomal protein (RPLPO) (See Methods). Graphed data reflect the mean ± S.D. of immunoreactivity and were analyzed with two-way ANOVA tests (Table 5) and post hoc Tukey tests. (DFn, DFd = 1, 40 for interaction; 1, 40 for sex; 1, 40 for exposure-ethanol/protein). Significant P-Values are indicated above the bars. * *p* < 0.05; ** *p* < 0.01; *** *p* < 0.001; **** *p* < 0.0001; ns = not significant.

**Figure 6 biomolecules-12-00676-f006:**
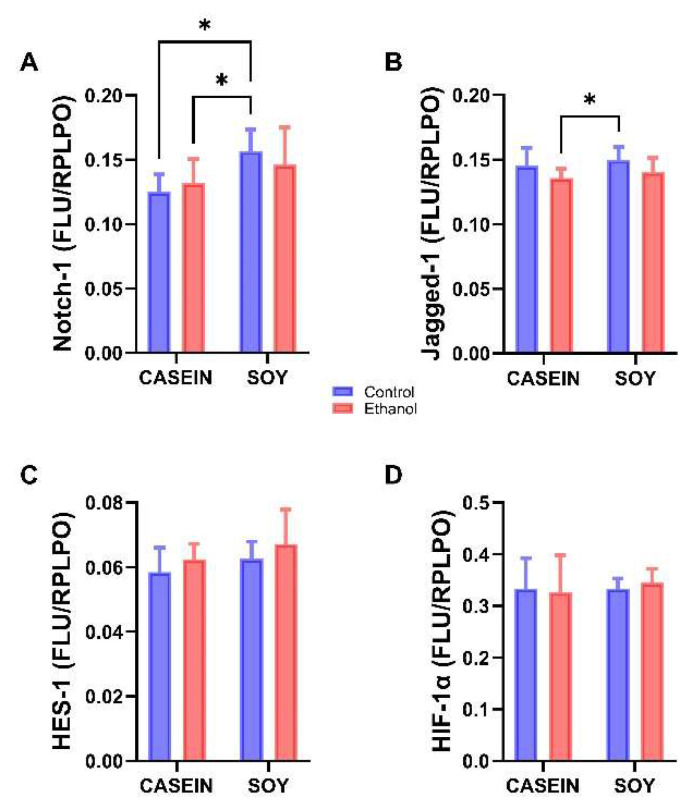
Effects of chronic ethanol and dietary soy isolate on Notch pathway protein expression. Duplex ELISAs were used to measure (**A**) Notch-1; (**B**) Jagged-1; (**C**) HES-1, and (**D**) HIF-1α (N = 12/group). Immunoreactivity was detected with horseradish peroxidase-conjugated secondary antibody and Amplex UltraRed soluble fluorophore. Results were normalized to large acidic ribosomal protein (RPLPO) (See Methods). Graphed data reflect the mean ± S.D. of immunoreactivity and were analyzed using two-way ANOVA tests (Table 5) with post hoc Tukey tests. Significant *p*-Values are indicated above the bars. * *p* < 0.05.

**Table 1 biomolecules-12-00676-t001:** Effects of Sex and Exposures (Ethanol + Dietary Protein) on Body and Brain Weight.

	Sex Effect		Exposure Effect		Sex x Exposure	
**Weight**	*f*-Ratio	*p*-Value	*f*-Ratio	*p*-Value	*f*-Ratio	*p*-Value
**Body**	*f* = 649.0	*p* < 0.0001	*f* = 214.2	*p* < 0.0001	*f* = 4.189	*p* = 0.011
**Brain**	*f* = 27.88	*p* < 0.0001	*f* = 19.34	*p* < 0.0001	*f* = 0.144	NS

Body weights were measured weekly. The mean area under curve calculations for body weight over the 7-week study were compared. Brain weights at the time of sacrifice were compared. The CC, EC, CS, and ES experimental groups each include 6 males and 6 females. For these two-way ANOVAS, exposures combined ethanol and dietary protein effects. Inter-group comparisons were made with a mixed-model ANOVA and post hoc Tukey tests with corrections for multiple comparisons (See Figure 1). *f*-Ratios correspond to sex effects, dietary protein + ethanol exposures, and interactive sex–exposure effects (DFn, DFd = 3, 48 for interaction; 1, 48 for sex; 3, 48 for exposure-ethanol/protein). NS = not significant. See Figure 1 for significant post hoc test results.

**Table 2 biomolecules-12-00676-t002:** Ethanol and Dietary Protein Effects on Morris Water Maze Performance.

	Trial Day Effect		Dietary Protein Effect		Trial Day x Diet	
**MWM**	*f*-Ratio	*p*-Value	*f*-Ratio	*p*-Value	*f*-Ratio	*p*-Value
**Latency**	*f* = 19.48	*p* < 0.0001	*f* = 4.259	*p* = 0.009	*f* = 0.723	NS
**Velocity**	*f* = 01.94	NS	*f* = 0.551	NS	*f* = 1.822	NS

Morris Water Maze (MWM) performance corresponding to the mean area-under-curve calculations of latencies to locate and land on the platform and swim velocities were compared over 4 consecutive trial days by two-way ANOVA. Since gender differences were not significant, male and female data were combined. *f*-Ratios correspond to Trial Day effects, dietary protein + ethanol exposure effects, and Trial Day x exposure interactive effects. NS = not significant. Graphs and post hoc Tukey test results are shown in Figure 2.

**Table 3 biomolecules-12-00676-t003:** Ethanol and Dietary Protein Effects on Akt-Pathway Proteins.

	Ethanol Effect		Diet Effect		Ethanol x Diet	
**Protein**	*f*-Ratio	*p*-Value	*f*-Ratio	*p*-Value	*f*-Ratio	*p*-Value
**IN-R**	1.128	NS	8.284	0.0093	10.13	0.0047
**IGF-1R**	1.194	NS	0.013	NS	3.887	0.063
**IRS-1**	0.002	NS	0.967	NS	6.08	0.0228
**Akt**	0.664	NS	1.356	NS	7.228	0.014
**GSK-3β**	0.100	NS	5.96	0.024	3.713	0.068
**PRAS40**	0.035	NS	1.961	NS	0.001	NS
**P70S6K**	0.379	NS	0.509	NS	5.402	0.031

Immunoreactivity was measured in 100 µg temporal lobe protein using a commercial 7-Plex Total Akt Magnetic ELISA panel. Immunoreactivity was measured with an MAGPIX instrument. Inter-group comparisons were made by mixed-model ANOVA and post hoc Tukey tests with correction for multiple comparisons (See Figure 3). F-Ratios correspond to measured ethanol effects, dietary soy effects or ethanol x Diet interactive effects. (DFn, DFd = 1, 40 for interaction; 1, 40 for sex; 1, 40 for exposure-ethanol/protein). Italicized p-values reflect statistical trends. NS = not significant.

**Table 4 biomolecules-12-00676-t004:** Ethanol and Dietary Protein Effects on Phospho-Akt-Pathway Proteins.

	Ethanol Effect		Diet Effect		Ethanol x Diet	
**Protein**	*f*-Ratio	*p*-Value	*f*-Ratio	*p*-Value	*f*-Ratio	*p*-Value
**pIN-R**	0.448	NS	4.975	0.037	3.184	0.089
**pIGF-1R**	2.422	NS	0.388	NS	1.302	NS
**pIRS-1**	0.427	NS	4.800	0.041	0.001	NS
**pAkt**	1.049	NS	0.485	NS	9.570	0.006
**pGSK-3b**	0.620	NS	0.775	NS	8.521	0.008
**pPRAS40**	0.521	NS	10.68	0.004	0.225	NS
**pP70S6K**	0.038	NS	0.000	NS	1.390	NS

Immunoreactivity was measured in 100 µg temporal lobe protein using a commercial 7-Plex Phospho-Akt Magnetic ELISA panel that detected ^pYpY1162/1163^-IR (pIN-R), ^pYpY1135/1136^-IGF-1R (pIGF-1R), ^pS473^-Akt (pAkt), ^pS9^-GSK-3β (pGSK-3β), ^pS312^-IRS-1 (pIRS-1), ^pT246^-PRAS40 (pPRAS40), and ^pTpS421/424^-p70S6K (pP70S6K). Immunoreactivity was measured with a MAGPIX instrument. Inter-group comparisons were made with a mixed-model ANOVA and post hoc Tukey test with corrections for multiple comparisons (See Figure 4). *f*-Ratios correspond to measured ethanol effects, dietary soy effects or ethanol x soy interactive effects. (DFn, DFd = 1, 40 for interaction; 1, 40 for sex; 1, 40 for exposure-ethanol/protein). Italicized *p*-values reflect statistical trends. NS = not significant.

**Table 5 biomolecules-12-00676-t005:** Ethanol and Dietary Protein Effects on ASPH and Notch Pathway Proteins.

	Ethanol Effect		Diet Effect		Ethanol x Diet	
**Protein**	*f*-Ratio	*p*-Value	*f*-Ratio	*p*-Value	*f*-Ratio	*p*-Value
**ASPH-A86G6**	*f* = 11.88	*p* = 0.003	*f* = 4.63	*p* = 0.044	*f* = 0.573	NS
**ASPH-A85E6**	*f* = 25.75	*p* < 0.0001	*f* = 15.02	*p* = 0.0009	*f* = 2.952	NS
**Notch-1**	*f* = 0.052	NS	*f* = 7.67	*p* = 0.012	*f* = 1.104	NS
**Jagged-1**	*f* = 4.573	*p* = 0.04	*f* = 1.12	NS	*f* < 0.001	NS
**HES-1**	*f* = 2.017	NS	*f* = 2.16	NS	*f* = 0.011	NS
**HIF-1α**	*f* = 0.016	NS	*f* = 0.20	NS	*f* = 0.209	NS

Immunoreactivity to ASPH-A85G6, ASPH-A85E6, Notch-1, Jagged-1, HES-1, and HIF-1α was measured in 100 ng of protein from temporal lobe homogenates. Duplex ELISAs were configured to quantify immunoreactivity in targe proteins using specific unlabeled monoclonal antibody coupled with horseradish peroxidase-conjugated secondary antibody and the Amplex UltraRed soluble fluorophore. To correct for modest differences in sample loading, results were normalized to large acidic ribosomal protein (RPLPO) measured in the same well with biotinylated antibody, Streptavidin-conjugated alkaline phosphatase and the 4-Methylumbelliferyl phosphate (4-MUP) substrate. Fluorescence intensities (Amplex Red: Ex 560 nm/Em 590 nm; 4-MUP: Ex 360/Em 450) were measured with a SpectraMax M5. Inter-group comparisons were made using mixed-model ANOVA and post hoc Tukey tests with correction for multiple comparisons (See Figure 5 and Figure 6). *f*-Ratios correspond to the measured ethanol effects, dietary protein effects or ethanol x dietary protein interactive effects. NS = not significant. (DFn, DFd = 1, 40 for interaction; 1, 40 for sex; 1, 40 for exposure-ethanol/protein).

## Data Availability

The data presented in this study are available upon request from Suzanne_DeLaMonte_MD@Brown.edu.

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
