# Peer review of "Dietary Soy Prevents Alcohol-Mediated Neurocognitive Dysfunction and Associated Impairments in Brain Insulin Pathway Signaling in an Adolescent Rat Model"

_biomolecules, 2022, doi:10.3390/biom12050676_

Round 1

Reviewer 1 Report

Comments- Accept with minor revisions

Authors Tong et al. discussed the findings of their research in the manuscript titled ' Dietary Soy Prevents Alcohol-Mediated Neurocognitive Dysfunction and Associated Impairments in Brain Insulin Pathway  Signaling in an Adolescent Rat Model'. Authors studied the effects of substituting soy isolate for casein to prevent or reduce ethanol’s adverse effects on brain structure and function. The research findings revealed that the insulin sensitizer properties of dietary soy isolate provide ample protection against several important long-term adverse effects of ethanol on neurobehavioral function and metabolic pathways in adolescent brains.

The authors have well written the current research article about the effects of consumption of dietary soy which prevents alcohol-mediated neurocognitive dysfunction. However, the reviewer has few minor concerns or suggestion for the authors.

  1. Authors have mentioned about the effect of dietary soy and ethanol + soy  on Phospho-Akt-Pathway Proteins, ASPH Proteins and Notch Pathway Proteins which were related to insulin signaling pathway. However, they have not mentioned any effect of ethanol and ethanol + soy on insulin level. Could you please mention the insulin level in mice before and after the treatments. 

Author Response

The authors thank this reviewer for the thoughtful comments that help to enhance understanding of how dietary soy positively impacts alcohol-related brain disease/degeneration. In response to the query about serum insulin levels, we measured insulin immunoreactivity in the serum samples using a gut hormone multiplex ELISA platform. The serum levels of insulin did not differ significantly among the groups or in relation to sex. This result is not surprising since the main actions of dietary soy are to function as insulin sensitizers and anti-oxidant/anti-inflammatory agents, reducing insulin resistance rather than insulin production.

We have added these comments to the Discussion.

Reviewer 2 Report

Title: Dietary Soy Prevents Alcohol-Mediated Neurocognitive Dys-2 function and Associated Impairments in Brain Insulin Pathway 3 Signaling in an Adolescent Rat Model

Abstract:

-The word “Methods” shouldn’t be underlined.

-“Morris Water maze test” instead of  “Morris Water maze tests”

-Results: Authors should be more specific when they refer to the results. For example, did casein reverse the ethanol-associated impairments in learning and memory? In the current form, the direction of the results is not evident. This is the case for brain atrophy, insulin signaling through Akt, and ASPH expression.

- Methods:  Important information is missing: For example, the type and dose of ethanol and casein treatment should be added, as well as the age of rats at the time of behavioral testing and sacrifice.

-Conclusions: Authors report that their treatment “.. provide ample protection against several important long-term adverse 34 effects of ethanol..”. The expression “long-term adverse effects” imply that behavioral testing took place some time following experimental manipulations. However, in the main text (p. 3) it appears that Morris WM was administered during the final week of experimental manipulation, which implies that the effects reported are not long-term.

-The originality of the study should be underlined.

Introduction

-p.2, l. 67-82: Authors should be more specific when they refer to the aim of the study. They need to present the novelty of their work and report the new questions to be tested. The gap in knowledge must be identified and the hypotheses need to be clearly stated.

 Materials and Methods

-p. 2-3: Important methodological information is missing: Authors should explain how the rats of different litters were allocated  to the different experimental groups. They should also report the route of administration of ethanol and soy isolate and their concentration.

-p. 3., l. 103: a)  it is not clear to the reviewer why MWM was administered during and  not following experimental manipulation. B) In addition, authors should report some technical characteristics of the maze such as diameter and height of walls, how many centimeters below the water surface was the platform submerged, and what time of the day behavioral testing took place? c) Authors have to report velocity data since latency is used as the dependent variable. d) Why authors administered MWM only 2 days given that in all standard protocols testing lasts at least 4 days with 4 daily trials? e) In the text it is reported that MWM was used to test spatial learning and memory. However, a probe trial was not administered, so how was spatial memory evaluated?

Results

-Body weight data are presented but authors haven’t referred at all at his variable in the methods section.

-Presentation of statistics is not appropriate. Statistical values need to be included in the text and, in addition to p values,  authors must report F values (with degrees of freedom).

-In figure legends, it should be added to which comparisons the various symbols refer.

-Authors claim that two-way ANOVAs were run. However, the way they analyzed the pairwise comparisons implies that they ran one-way Anovas followed by post-hoc. This statistical analysis wouldn’t be appropriate given that their design is factorial.

Discussion

- In the current form, discussion is poorly written. Authors should try to connect their findings with existing evidence. It should be clear to the reader how their results fit in with current knowledge. In addition, authors need to explain contradictory findings.

Author Response

The authors thank this reviewer for the thoughtful comments that help to enhance understanding of how dietary soy positively impacts alcohol-related brain disease/degeneration.

Our point-by-point responses are in purple font. The same information has been uploaded.

Reviewer #2:

Title
: Dietary Soy Prevents Alcohol-Mediated Neurocognitive Dys-2 function and Associated Impairments in Brain Insulin Pathway 3 Signaling in an Adolescent Rat Model

Abstract:

-The word “Methods” shouldn’t be underlined.

            Underline removed

-“Morris Water maze test” instead of  “Morris Water maze tests”

            Plural removed

-Results: Authors should be more specific when they refer to the results. For example, did casein reverse the ethanol-associated impairments in learning and memory? In the current form, the direction of the results is not evident. This is the case for brain atrophy, insulin signaling through Akt, and ASPH expression.

            The Results section of the Abstract has been clarified to indicate that the adverse effects of ethanol observed when standard casein is used as the protein source were prevented by replacing casein with dietary soy.

- Methods:  Important information is missing: For example, the type and dose of ethanol and casein treatment should be added, as well as the age of rats at the time of behavioral testing and sacrifice.

            Accordingly, we have edited the methods section of the Abstract.

-Conclusions: Authors report that their treatment “.. provide ample protection against several important long-term adverse 34 effects of ethanol..”. The expression “long-term adverse effects” imply that behavioral testing took place some time following experimental manipulations. However, in the main text (p. 3) it appears that Morris WM was administered during the final week of experimental manipulation, which implies that the effects reported are not long-term.

            The Morris WM studies were done in rats after 7 weeks of ethanol  feeding. Previous studies showed that 5-6 weeks of chronic ethanol feeding is sufficient to generate a model of alcohol-related brain degeneration. The goal was to determine if soy isolate substituted for casein would prevent development of alcohol-related brain degeneration. However, we did not determine what would happen if the dietary soy were to be discontinued. We have corrected the statement and focus on prevention or reduction, and omitted the concept of long-term adverse effects.

-The originality of the study should be underlined.

            We have underlined the Conclusion statement as that reflects the originality of the work.

Introduction

-p.2, l. 67-82: Authors should be more specific when they refer to the aim of the study. They need to present the novelty of their work and report the new questions to be tested. The gap in knowledge must be identified and the hypotheses need to be clearly stated.

            We have strengthened the comments pertaining to the aims of the study and how the research addresses current knowledge gaps [pg3, li 83-92].

 Materials and Methods

-p. 2-3: Important methodological information is missing: Authors should explain how the rats of different litters were allocated to the different experimental groups. They should also report the route of administration of ethanol and soy isolate and their concentration.

            The requested details are now included—Page 3, Lines 102-116.

-p. 3., l. 103: a)  it is not clear to the reviewer why MWM was administered during and  not following experimental manipulation. B) In addition, authors should report some technical characteristics of the maze such as diameter and height of walls, how many centimeters below the water surface was the platform submerged, and what time of the day behavioral testing took place? c) Authors have to report velocity data since latency is used as the dependent variable. d) Why authors administered MWM only 2 days given that in all standard protocols testing lasts at least 4 days with 4 daily trials? e) In the text it is reported that MWM was used to test spatial learning and memory. However, a probe trial was not administered, so how was spatial memory evaluated?

            The experimental timeline was that beginning at 4 weeks of age, the rats were maintained on the liquid diets for 7 weeks. During the final week of feeding, the rats were subjected to MWM testing over a period of 4 consecutive days. It was critical to continue the diets throughout the study to avoid ethanol withdrawal effects. The MWM tests were run over 4 days not 2, and with 3 daily trials. Correct, we did not include a probe trial. Velocity data are shown in Supplementary Figure 1 (Figure A1) as there were no inter-group differences.

Results

-Body weight data are presented but authors haven’t referred at all at his variable in the methods section.

            A reference to weekly body weights was included in the Methods section “Rats were monitored daily and weighed weekly to ensure adequate food intake and growth.” Pg 3, Lines 115-116, and under Results, Pg 6, Line 215.

-Presentation of statistics is not appropriate. Statistical values need to be included in the text and, in addition to p values,  authors must report F values (with degrees of freedom).

            All of the statistical values are tabulated. The F-ratios and p-values are listed in the tables. DF’s have been added to the table legends. It would be extremely cumbersome to put 5 Tables worth of statistics in the main text.

-In figure legends, it should be added to which comparisons the various symbols refer.

            The symbols related to inter-group comparisons are now explained in the Figure legends.

-Authors claim that two-way ANOVAs were run. However, the way they analyzed the pairwise comparisons implies that they ran one-way Anovas followed by post-hoc. This statistical analysis wouldn’t be appropriate given that their design is factorial.

            Indeed, two-way ANOVA tests were run. We noted that some of the tables have the column heading that indicates interactive effects, e.g. ethanol x soy, cut off. That has been repaired. One-way ANOVA tests were not used. We have repositioned the tables to show all data.

Discussion

- In the current form, discussion is poorly written. Authors should try to connect their findings with existing evidence. It should be clear to the reader how their results fit in with current knowledge. In addition, authors need to explain contradictory findings.

We have revised the Discussion and had it read by others for clarity. The dietary soy study is novel. There is no prior data showing benefits of dietary soy on brain function in an adolescent model of chronic alcohol exposure.

We were able to demonstrate again that chronic alcohol exposure causes brain atrophy and impairs brain insulin signaling networks through PI3K-Akt pathways. The rescue effects of dietary soy were not previously known However, we have published similar therapeutic effects of insulin sensitizers in related models.

Although the responses to ethanol were less pronounced than in other studies, we modified the model by excluding superimposed binge exposures, which generate a very robust model. However, in a pilot study we found that females were more sensitive to ethanol than males. To maintain good health of both sexes, we elected to keep the ethanol dosing moderately high rather than maximum  and omit the binge exposures. These points are now included in the Discussion [pg 17, Li 376-386].

Round 2

Reviewer 2 Report

Response of the reviewer to the authors’ changes appear in red

Reviewer #2:

Title
: Dietary Soy Prevents Alcohol-Mediated Neurocognitive Dys-2 function and Associated Impairments in Brain Insulin Pathway 3 Signaling in an Adolescent Rat Model

Abstract:

-The word “Methods” shouldn’t be underlined.

            Underline removed

-“Morris Water maze test” instead of  “Morris Water maze tests”

            Plural removed

**Comments have been addressed.

-Results: Authors should be more specific when they refer to the results. For example, did casein reverse the ethanol-associated impairments in learning and memory? In the current form, the direction of the results is not evident. This is the case for brain atrophy, insulin signaling through Akt, and ASPH expression.

            The Results section of the Abstract has been clarified to indicate that the adverse effects of ethanol observed when standard casein is used as the protein source were prevented by replacing casein with dietary soy.

**The expression “inhibitory effects on brain weight” is not specific. It should be clearly stated that brain weight was reduced.

- Methods:  Important information is missing: For example, the type and dose of ethanol and casein treatment should be added, as well as the age of rats at the time of behavioral testing and sacrifice.

            Accordingly, we have edited the methods section of the Abstract.

**Comment has been addressed.

-Conclusions: Authors report that their treatment “.. provide ample protection against several important long-term adverse 34 effects of ethanol..”. The expression “long-term adverse effects” imply that behavioral testing took place some time following experimental manipulations. However, in the main text (p. 3) it appears that Morris WM was administered during the final week of experimental manipulation, which implies that the effects reported are not long-term.

            The Morris WM studies were done in rats after 7 weeks of ethanol  feeding. Previous studies showed that 5-6 weeks of chronic ethanol feeding is sufficient to generate a model of alcohol-related brain degeneration. The goal was to determine if soy isolate substituted for casein would prevent development of alcohol-related brain degeneration. However, we did not determine what would happen if the dietary soy were to be discontinued. We have corrected the statement and focus on prevention or reduction, and omitted the concept of long-term adverse effects.

** Comment has been addressed.

-The originality of the study should be underlined.

            We have underlined the Conclusion statement as that reflects the originality of the work.

** Added information in the conclusion section is very helpful. However, the originality of the study should be also stressed in the introduction.

Introduction

-p.2, l. 67-82: Authors should be more specific when they refer to the aim of the study. They need to present the novelty of their work and report the new questions to be tested. The gap in knowledge must be identified and the hypotheses need to be clearly stated.

            We have strengthened the comments pertaining to the aims of the study and how the research addresses current knowledge gaps [pg3, li 83-92].

**Hypotheses are still not clearly stated.

 Materials and Methods

-p. 2-3: Important methodological information is missing: Authors should explain how the rats of different litters were allocated to the different experimental groups. They should also report the route of administration of ethanol and soy isolate and their concentration.

            The requested details are now included—Page 3, Lines 102-116.

**Authors should have not explained how the rats of different litters were allocated to the different experimental groups. For example, from how many litters rats of each group originated?

-p. 3., l. 103: a)  it is not clear to the reviewer why MWM was administered during and  not following experimental manipulation. B) In addition, authors should report some technical characteristics of the maze such as diameter and height of walls, how many centimeters below the water surface was the platform submerged, and what time of the day behavioral testing took place? c) Authors have to report velocity data since latency is used as the dependent variable. d) Why authors administered MWM only 2 days given that in all standard protocols testing lasts at least 4 days with 4 daily trials? e) In the text it is reported that MWM was used to test spatial learning and memory. However, a probe trial was not administered, so how was spatial memory evaluated?

            The experimental timeline was that beginning at 4 weeks of age, the rats were maintained on the liquid diets for 7 weeks. During the final week of feeding, the rats were subjected to MWM testing over a period of 4 consecutive days. It was critical to continue the diets throughout the study to avoid ethanol withdrawal effects. The MWM tests were run over 4 days not 2, and with 3 daily trials. Correct, we did not include a probe trial. Velocity data are shown in Supplementary Figure 1 (Figure A1) as there were no inter-group differences.

** 1. Although authors report that tests were performed between 10 AM and 4 PM, they should add the dark and light phases of the light/dark cycle in section 2.1, so that it would be clear whether testing took place during light or dark phase.

2. The caption text of Fig. A1 should be limited to data being presented (i.e., velocity) and should appear as Fig 2A and not as “Supplementary Figure”. In addition, statistics about velocity should be added in section 3.2.

Results

-Body weight data are presented but authors haven’t referred at all at his variable in the methods section.

            A reference to weekly body weights was included in the Methods section “Rats were monitored daily and weighed weekly to ensure adequate food intake and growth.” Pg 3, Lines 115-116, and under Results, Pg 6, Line 215.

**Comments have been addressed.

-Presentation of statistics is not appropriate. Statistical values need to be included in the text and, in addition to p values,  authors must report F values (with degrees of freedom).

            All of the statistical values are tabulated. The F-ratios and p-values are listed in the tables. DF’s have been added to the table legends. It would be extremely cumbersome to put 5 Tables worth of statistics in the main text.

**Comments have not been addressed.

-In figure legends, it should be added to which comparisons the various symbols refer.

            The symbols related to inter-group comparisons are now explained in the Figure legends.

**Comment has been addressed.

-Authors claim that two-way ANOVAs were run. However, the way they analyzed the pairwise comparisons implies that they ran one-way Anovas followed by post-hoc. This statistical analysis wouldn’t be appropriate given that their design is factorial.

            Indeed, two-way ANOVA tests were run. We noted that some of the tables have the column heading that indicates interactive effects, e.g. ethanol x soy, cut off. That has been repaired. One-way ANOVA tests were not used. We have repositioned the tables to show all data.

**Comment has been addressed.

Discussion

- In the current form, discussion is poorly written. Authors should try to connect their findings with existing evidence. It should be clear to the reader how their results fit in with current knowledge. In addition, authors need to explain contradictory findings.

We have revised the Discussion and had it read by others for clarity. The dietary soy study is novel. There is no prior data showing benefits of dietary soy on brain function in an adolescent model of chronic alcohol exposure.

We were able to demonstrate again that chronic alcohol exposure causes brain atrophy and impairs brain insulin signaling networks through PI3K-Akt pathways. The rescue effects of dietary soy were not previously known However, we have published similar therapeutic effects of insulin sensitizers in related models.

Although the responses to ethanol were less pronounced than in other studies, we modified the model by excluding superimposed binge exposures, which generate a very robust model. However, in a pilot study we found that females were more sensitive to ethanol than males. To maintain good health of both sexes, we elected to keep the ethanol dosing moderately high rather than maximum  and omit the binge exposures. These points are now included in the Discussion [pg 17, Li 376-386].

**Comment has been addressed.

Author Response

Below is the same content provided in the uploaded file.

Response of the reviewer to the authors’ changes appear in red

Authors’ changes made in response to Review #2 are in Blue font

Reviewer #2:

Title
: Dietary Soy Prevents Alcohol-Mediated Neurocognitive Dys-2 function and Associated Impairments in Brain Insulin Pathway 3 Signaling in an Adolescent Rat Model

Abstract:

-Results: Authors should be more specific when they refer to the results. For example, did casein reverse the ethanol-associated impairments in learning and memory? In the current form, the direction of the results is not evident. This is the case for brain atrophy, insulin signaling through Akt, and ASPH expression.

            The Results section of the Abstract has been clarified to indicate that the adverse effects of ethanol observed when standard casein is used as the protein source were prevented by replacing casein with dietary soy.

**The expression “inhibitory effects on brain weight” is not specific. It should be clearly stated that brain weight was reduced.

The first sentence in the Results section of the Abstract was revised to make it declarative

-The originality of the study should be underlined.

            We have underlined the Conclusion statement as that reflects the originality of the work.

** Added information in the conclusion section is very helpful. However, the originality of the study should be also stressed in the introduction.

The originality of the study has been emphasized under Introduction, Page 3, Lines 96-97

Introduction

-p.2, l. 67-82: Authors should be more specific when they refer to the aim of the study. They need to present the novelty of their work and report the new questions to be tested. The gap in knowledge must be identified and the hypotheses need to be clearly stated.

            We have strengthened the comments pertaining to the aims of the study and how the research addresses current knowledge gaps [pg3, li 83-92].

**Hypotheses are still not clearly stated.

The hypothesis has been re-stated for clarity and specificity, Page 3, Lines 84-87.

 Materials and Methods

-p. 2-3: Important methodological information is missing: Authors should explain how the rats of different litters were allocated to the different experimental groups. They should also report the route of administration of ethanol and soy isolate and their concentration.

            The requested details are now included—Page 3, Lines 102-116.

**Authors should have not explained how the rats of different litters were allocated to the different experimental groups. For example, from how many litters rats of each group originated?

We have specified that the rats were drawn from 7 different liters, Page 3, Line 109.

-p. 3., l. 103: a)  it is not clear to the reviewer why MWM was administered during and  not following experimental manipulation. B) In addition, authors should report some technical characteristics of the maze such as diameter and height of walls, how many centimeters below the water surface was the platform submerged, and what time of the day behavioral testing took place? c) Authors have to report velocity data since latency is used as the dependent variable. d) Why authors administered MWM only 2 days given that in all standard protocols testing lasts at least 4 days with 4 daily trials? e) In the text it is reported that MWM was used to test spatial learning and memory. However, a probe trial was not administered, so how was spatial memory evaluated?

            The experimental timeline was that beginning at 4 weeks of age, the rats were maintained on the liquid diets for 7 weeks. During the final week of feeding, the rats were subjected to MWM testing over a period of 4 consecutive days. It was critical to continue the diets throughout the study to avoid ethanol withdrawal effects. The MWM tests were run over 4 days not 2, and with 3 daily trials. Correct, we did not include a probe trial. Velocity data are shown in Supplementary Figure 1 (Figure A1) as there were no inter-group differences.

** 1. Although authors report that tests were performed between 10 AM and 4 PM, they should add the dark and light phases of the light/dark cycle in section 2.1, so that it would be clear whether testing took place during light or dark phase.

The dark and light phases are now indicated, Page 3, Lines 103-104.

2. The caption text of Fig. A1 should be limited to data being presented (i.e., velocity) and should appear as Fig 2A and not as “Supplementary Figure”. In addition, statistics about velocity should be added in section 3.2.

Supplementary Figure has been eliminated and instead, the Figure is now 2B. The excessive information in the legend has been omitted. Statistics corresponding to Figure 2B have been included in Table 2.

Results

**Comment has been addressed.

Discussion

**Comment has been addressed.